# Road lighting density and brightness linked with increased cycling rates after-dark

**Jim Uttley**[1]*, **Steve Fotios**[1], **Robin Lovelace**[2]

**1** School of Architecture, Arts Tower, University of Sheffield, Sheffield, England, United Kingdom, **2** Institute for Transport Studies, University of Leeds, Leeds, England, United Kingdom

* j.uttley@sheffield.ac.uk

**Data Availability Statement:** All relevant data are within the paper, its Supporting Information files, and the Open Science Framework database, at https://osf.io/w32qe/.

**Funding:** JU, SF and RL were supported by funding from the White Rose Collaboration Fund

## Abstract

Cycling has a range of benefits as is recognised by national and international policies aiming to increase cycling rates. Darkness acts as a barrier to people cycling, with fewer people cycling after-dark when seasonal and time-of-day factors are accounted for. This paper explores whether road lighting can reduce the negative impact of darkness on cycling rates. Changes in cycling rates between daylight and after-dark were quantified for 48 locations in Birmingham, United Kingdom, by calculating an odds ratio. These odds ratios were compared against two measures of road lighting at each location: 1) Density of road lighting lanterns; 2) Relative brightness as estimated from night-time aerial images. Locations with no road lighting showed a significantly greater reduction in cycling after-dark compared with locations that had some lighting. A nonlinear relationship was found between relative brightness at a location at night and the reduction in cyclists after-dark. Small initial increases in brightness resulted in large reductions in the difference between cyclist numbers in daylight and after-dark, but this effect reached a plateau as brightness increased. These results suggest only a minimal amount of lighting can promote cycling after-dark, making it an attractive mode of transport year-round.

## Introduction

Cycling has a range of health, economic and environmental benefits [1]. A mode shift in transport from personal motorised vehicles to cycling addresses three interrelated crises: climate change via reduced energy use [2]; air pollution through reductions in pollutants emitted by motor vehicles [3]; and premature deaths caused by physical inactivity [4]. More widely, increasing cycling can improve the physical and mental wellbeing of individuals and communities, helping reduce the annual £7.4 billion cost to the UK of physical inactivity [5]. Cycling as a mode of transport can also increase travel satisfaction, relative to other modes of transport such as public transit [6]. In recognition of these benefits, the UK Government has enacted a statutory Cycling and Walking Investment Strategy [7], the aim of which is to double the number of cycling trips by 2025, from a baseline level in 2013. Similar targets have been adopted in many countries and cities worldwide [8].

Demand for travel has historically clustered at certain times of the day, such as morning and evening rush hours associated with travel to and from work. In recent years, however,

(grant number 156259-1) - https://whiterose.ac.uk/collaboration-fund/ JU and SF were supported by funding from the MERLIN-2 project, funded by the Engineering and Physical Sciences Research Council (grant number EP/M02900X/1) - https://gow.epsrc.ukri.org/NGBOViewGrant.aspx?GrantRef=EP/M02900X/1 These funders had no role in study design, data collection and analysis, decision to publish, or preparation of the manuscript

**Competing interests:** The authors have declared no competing interests exist.

travel demand in modern society has dispersed throughout the 24-hour period as demand for services, working patterns and leisure activities spread across the entire day [9]. More businesses are operating around the clock as a result of globalisation and the introduction of new technologies that increase connectivity, such as mobile phones and the internet [10]. These changes are partly in response to increased demand for out-of-hours services, for example with more people wanting to shop outside traditional daytime hours [11]. The introduction of flexible working policies in many Government and business employers has also led to the diffusion of activities across the 24-hour period [12].

In this context it is important that sustainable travel options are available at all times of the day. However, environmental conditions change over a 24-hour period and these conditions may influence whether people decide to cycle or not. The most obvious diurnal variation in environmental conditions is the change in light levels, from daylight to varying degrees of darkness depending on the time of day and presence of artificial lighting. The amount of daylight and darkness in a day varies with the time of year, and is also a function of a location's latitude, with more extreme latitude locations having longer durations of darkness during parts of the year than countries at less extreme latitudes.

Being able to make a trip during daylight hours may be one of the most significant motivations for someone deciding whether to cycle. For example, respondents cited being able to make the trip in daylight hours as one of the top motivators when asked how different factors would influence the decision to cycle [13]. The impact of ambient light conditions on whether people choose to cycle or not is thus an increasingly important consideration in sustainable transport planning.

Recent work [14,15] has confirmed the link between ambient light and cycling rates using observational data (counts of cyclists) in a United States district. Daylight produced a relative increase in cyclists of between 38% and 67% compared with darkness in the same hour of the day, depending on the method of analysis used. The effect of ambient light on the number of pedestrians was even greater than that for cyclists. A case-control method was used in that work, with an odds ratio quantifying the effect of darkness on cycling rates, relative to daylight. The case-control method accounts for potentially confounding factors that influence cycling rates but are not related to ambient light conditions. For example, simply comparing counts of cyclists during daylight and darkness would not account for two of the most significant influences on cyclist numbers—the time of day and time of year [16]. Darkness is associated with evenings and early mornings, and winter. The case-control method accounts for daily and seasonal effects by comparing counts in the same hour of the day but in daylight or darkness (a 'case' hour), and contrasting with counts over the same time periods but in an hour where the ambient light remains constant (e.g. always daylight or always dark, regardless of time of year–a 'control' hour). This odds ratio method can be applied to counts of cyclists to estimate effects of ambient light on cycling rates, but it has also been used in investigations of the effect of ambient light on traffic collisions and pedestrian safety [17,18].

Outdoor lighting of cycle routes, such as road lighting, can reduce the negative impact of darkness on cycling rates. Winters et al [13] found that a major deterrent to cycling was if the route was not well lit after-dark. The deterrent effect was largest for people who don't currently cycle but potentially would do in the future, highlighting the importance of lighting in encouraging uptake of cycling. Recent research [19] found that only 23% of women felt safe cycling during hours of darkness, compared with 36% of men, which suggests that Lighting may also contribute to addressing the current gender imbalance in cycling by encouraging more women to cycle after-dark.

There are several reasons why lighting can encourage someone to cycle after-dark. Detecting and responding to potential hazards in the road or path, such as obstacles, potholes and

pedestrians, is a critical task for cyclists [20–22]. Lighting can improve the visibility of such hazards [23], giving a person greater confidence to cycle. Lighting can also reduce the potential of being hit by drivers [24], by improving visibility of the cyclist. This can also give greater confidence to cycle. Lighting is also associated with increased reassurance and reduced fear of crime [25]. People may therefore feel safer and less threatened by criminals when using lit roads after-dark, leading to more people willing to cycle.

Despite the potential for lighting to influence cycling rates, little research has been undertaken in this area. As part of their work examining the impact of ambient light on cycling rates, Uttley and Fotios [14] found that darkness produced a larger reduction in cyclists on off-road cycle trails compared with on-road cycle lanes. One hypothesis to explain this difference was potential differences in lighting at these two types of locations. This hypothesis was explored in further work [15], which found that 39% of counter locations on off-road cycle trails were unlit, whereas all on-road cycle lane locations were lit. Comparing only the locations on cycle trails, the effect of darkness was found to be significantly greater where there was no lighting present compared with locations that had some lighting.

A limitation of this past work is that the definition of ambient light conditions was rather coarse, i.e. either daylight or after dark, and for after-dark conditions, whether lit or unlit. For lit conditions there can be a wide range of light levels, with design guidance for road lighting recommending average illuminances from 2 lux on residential or subsidiary roads to 50 lux at major road intersections [26].

The current paper presents an analysis of the relationship between ambient light, lighting and cycling rates using more precise quantitative definitions of lighting than have previously been used. Two alternative measures of lighting are used—lantern density and relative brightness. Lantern density (the number of road lights on a standard length of road or path) may be linked to aspects of the light conditions experienced by a cyclist, such as illuminance and uniformity, and this metric was previously used in research about lighting and crime [27]. Relative brightness was established using night time aerial images, as used in previous research examining light pollution and its links to urban land use [28]. It has been shown that pixel values in night time aerial images are strongly associated with measurements of actual illuminances on the ground [29].

Our aim is to assess the relationship between characteristics of lighting and the rates of cycling after-dark. To address this aim, we assess four hypotheses:

1. Fewer people cycle when it is dark compared with daylight, when factors not related to ambient light are controlled for

2. Fewer people cycle when it is twilight compared with daylight, but the effect will not be as large as when it is fully dark

3. Higher lantern density will reduce the negative impact of darkness and twilight on cyclist numbers

4. More brightly lit locations will show a reduced impact of darkness and twilight on cyclist numbers, compared with less brightly lit locations

## Method

### Analytical strategy

We use lantern density and relative brightness (as estimated from aerial images) to characterise lighting at locations with cycle counters in a UK city. Lighting conditions are compared against the effect of darkness on cyclist numbers at these locations, with effectiveness quantified using

the odds ratio method. The effect of twilight conditions is also measured, to assess whether the magnitude of darkness is related to the impact on cyclist numbers.

Counts of cyclist numbers from automated cycle counters in Birmingham, UK, were obtained for each hour of the day over a 4-year period, 2012 to 2015. The odds ratio method (described below) was used to estimate the effect of twilight and darkness on cyclist numbers compared with daylight conditions. These odds ratios were compared against two measures of lighting at the analysed locations: 1) The density of public lighting columns / lanterns on a road; and 2) An estimate of road brightness, obtained from night time aerial photography images. These lighting measures were calculated for the stretch of road or path where the count location was situated.

## Measuring the effect of darkness—odds ratios

Previous research [14,15,17,18] has used odds ratios to quantify the effect of darkness on the number of people walking and cycling, and the risk to pedestrians at pedestrian crossings. The approach identifies a 'case' hour of the day that is in darkness for part of the year and daylight for the rest of the year. 'Control' hours are also identified, these having the same light condition throughout the year (always daylight, or always dark). The ratio of daylight and darkness cyclist counts in the case hour is compared to the ratio calculated over the same periods but in the control hour. Comparing the ratio in the case hour with that for the control hour(s) helps to account for factors that may change over the course of the year but not be directly caused by changes in light conditions, such as the weather. The large changes in daylight length across the year mean a case hour can be selected that will provide starkly different ambient light conditions at different times of the year.

An alternative approach to comparing counts across a whole year would be to compare counts in the weeks immediately before and after a daylight saving time clock change (e.g. [14]). This approach uses the clock change to produce a rapid transition between daylight and dark conditions within an appropriately selected case hour. However, we chose to use the whole year approach to increase the sample of cyclist counts we were using. This was important as we were not only calculating an overall odds ratio for the effect of ambient light condition, we were also calculating individual odds ratios for each counter location. Such disaggregation of the data required as large a sample as possible, hence the use of the whole year method rather than the clock change method.

The case hour was chosen as 18:00–18:59. The hours of 14:00–14:59 (daylight throughout year) and 22:00–22:59 (dark throughout year) were selected as control hours. Two control hours were used in order to provide both ambient light conditions, and to capture different types of cyclist who may cycle at different times of the day. Previous analysis has suggested the time of a control hour can influence the magnitude of a calculated odds ratio, although for cyclists this influence is relatively small [14]. The choice of hours selected for the control period was based on the need to compare consistent ambient light conditions throughout the year. The hour beginning at 22:00 was the earliest hour that was after sunset throughout the year, even at Summer solstice; 14:00 was the latest hour that was always before sunset throughout the year, even at Winter solstice. These hours were also equally spaced either side of the case hour–four hours before or after.

We only selected an evening case hour as in a morning case hour (e.g. 06:00–07:00) the light level will fluctuate in a non-continuous way around the biannual clock changes that occur in the UK (and many other countries). Clocks go forwards one hour in Spring and back one hour in Autumn. In the evening, this only emphasises the gradual transition from darkness to daylight in the Spring, and from daylight to darkness in Autumn. In the morning

however, the clock change results in a brief reversal of light conditions. This means a morning case hour does not undergo a continuous transition through the different light conditions, as an evening case hour does, but undergoes a disjointed set of light conditions. This adds a layer of complexity to the analysis, and may introduce a mediating factor that adds noise to results related to the effects of light conditions on cycling rates. See also the work of Sullivan and Flannagan [30], in which they also only examine light transitions in the evening to assess the impact of darkness on collisions involving pedestrians.

Counts in both these control hours were combined to average out these small variations depending on choice of control hour, increase the sample of cyclist counts, and simplify reporting of results.

Table 1 summarises the ambient light condition for each of these hours during different periods of the year. For short periods of the year the case hour was in a transitional light condition, for example partly in darkness and partly in twilight, or partly in twilight and partly in daylight. The predominant light condition (i.e. that applied to more than half of the hour) was assigned to those dates with transitional light conditions. Daylight was defined as the period before sunset; twilight defined as the period between sunset and the end of civil twilight; and darkness defined as the period after civil twilight. The times for sunset and civil twilight for Birmingham, UK, were taken from the Time and Date website [31].

Eq 1 was used to calculate an odds ratio, showing the effect of darkness/twilight on cyclist frequency, for each counter location.

$$Odds\ ratio_i = \frac{Case_{day,\ i} \big/ Case_{dark,\ i}}{Control_{day,\ i} \big/ Control_{dark,\ i}} \tag{1}$$

Where, for counter location $i$, $Case_{dark}$ is the count of cyclists in the case hour when it is dark (or twilight); $Case_{day}$ is the count of cyclists in the case hour when it is in daylight; $Control_{dark}$ is the count of cyclists in the control hours when the case hour is dark (or twilight); and $Control_{day}$ is the count of cyclists in the control hours when the case hour is in daylight.

## Cyclist count data

Open source cyclist count data were downloaded from the Birmingham Data Factory website [32] for the 4-year period between 01/01/2012 to 31/12/2015 inclusive (all months in each year were included in the dataset). These data provided hourly counts per day at 48 locations within the Birmingham Local Authority district of the United Kingdom. One additional counter (CY52) was excluded because lighting data are not available—it is located in an underpass. Fig 1 shows the Birmingham district boundary and locations of the 48 cycle counters (see also S1 Dataset). These counters became active at different times between 2012 and 2015. The majority (90%) were commissioned during 2012, with the remaining counters commissioned during 2015. The months and years of commission for each counter are shown in Table 2.

**Table 1. Light condition of case and control hours during different periods of the year.**

| Dates | Total number of days | Case hour (18:00–18:59) | Control hour 1 (14:00–14:59) | Control hour 2 (22:00–22:59) |
|---|---|---|---|---|
| 1 January - 6 March | 65 | Darkness | Daylight | Darkness |
| 7 March - 25 March | 19 | Twilight | Daylight | Darkness |
| 26 March - 7 October | 196 | Daylight | Daylight | Darkness |
| 8 October - 23 October | 16 | Twilight | Daylight | Darkness |
| 24 October - 31 December | 70 | Darkness | Daylight | Darkness |

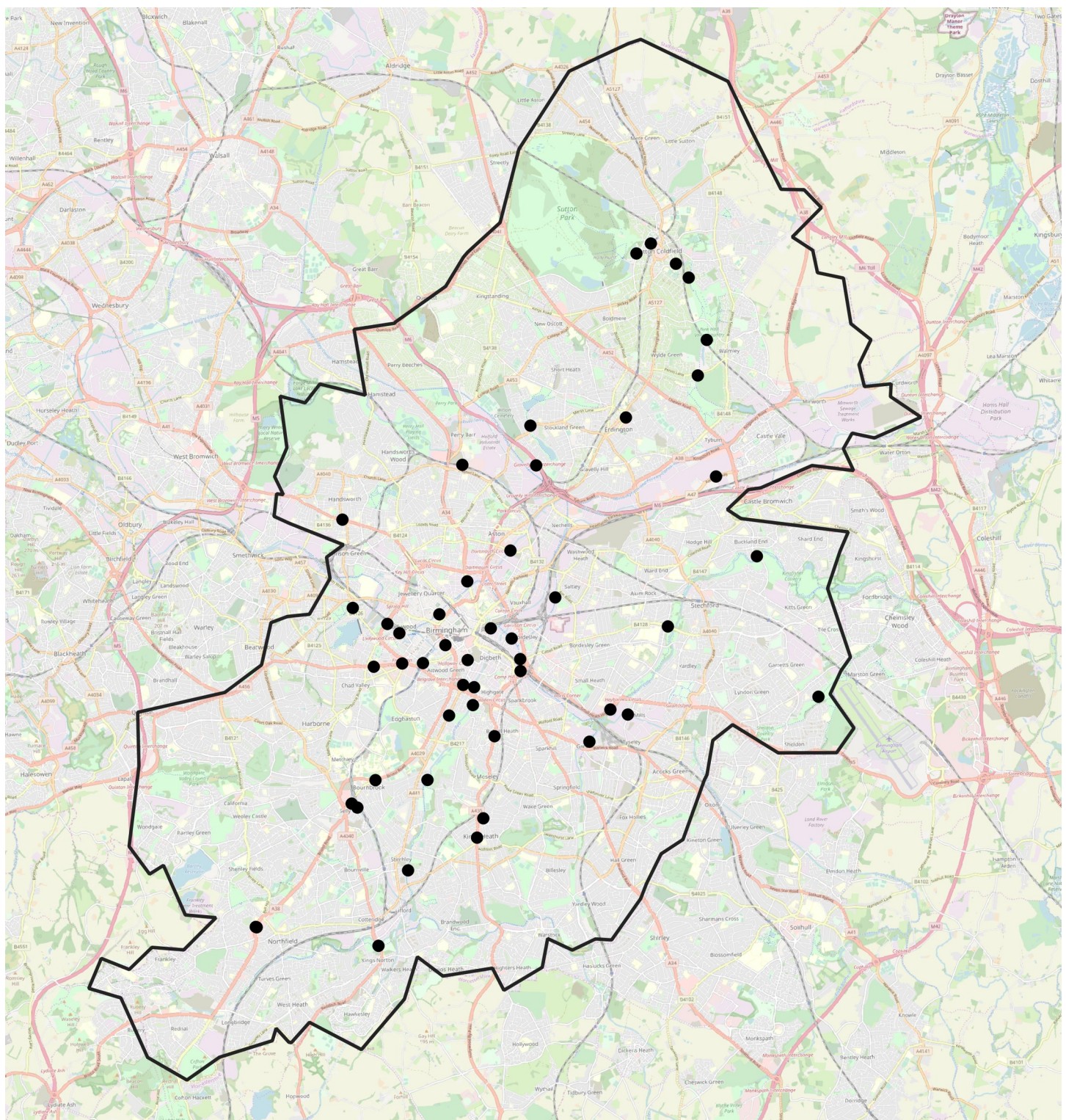

**Fig 1. Boundary of Birmingham Local Authority district, within UK, and locations of 48 cycle counters.** Map imagery from OpenStreetMap Humanitarian map style, CC0 (public domain) license, https://github.com/hotosm/HDM-CartoCSS/blob/master/LICENCE.txt.

**Table 2. Month and year of commission for the Birmingham cycle counters.**

| Month and year of commission | Cycle Counter Identity Numbers[a] |
|---|---|
| January 2012 | CY11; CY13; CY15; CY16; CY17; CY18 |
| March 2012 | CY53N; CY53S; CY54; CY55; CY56; CY57; CY58 |
| April 2012 | CY71 |
| May 2012 | CY59; CY60; CY69; CY73; CY76; CY77; CY78 |
| June 2012 | CY65; CY68; CY70; CY72 |
| July 2012 | CY64; CY66; CY67; CY74; CY75; CY79; CY80; CY83 |
| August 2012 | CY12; CY47; CY49; CY81; CY82; CY90; CY91 |
| September 2012 | CY48 |
| October 2012 | CY46; CY92 |
| July 2015 | CY95; CY96; CY97; CY98 |
| August 2015 | CY99 |

[a] Cycle counter identity numbers refer to the original codes used in the open-source data, available from Birmingham City Council (2019).

The downloaded data provided counts by the hour for two directions (e.g. Northbound and Southbound, or Eastbound and Westbound) at each counter location. Data were filtered to include only counts during the case hour (18:00–18:59) and the control hours (14:00–14:59 and 22:00–22:59). Any days that did not provide data from a counter for all three hours (e.g. as a result of the counter being out of operation due to damage or maintenance) were excluded from further analysis. This resulted in the exclusion of 1% of the total amount of days that the counters were commissioned for, between 2012 and 2015. The remaining data were summed across both directions for each counter. Data were then aggregated to provide total counts for each counter in the case hour and control hours, by year and by the light condition during the case hour (these data are available in S2 and S3 Datasets). Odds ratios were calculated for each counter location using Eq 1.

## Road network

The goal of the current work was to compare the effect of darkness on cyclist counts at each counter location, as measured by the counter's odds ratio, against measures of lighting at those locations. These lighting measures were calculated for the stretch of road the counter was located on. The Ordnance Survey OpenRoads product provided data about road locations within Birmingham (available for open source download, Ordnance Survey (2018)), and the road network for Birmingham was plotted using the GIS software QGIS (version 2.18, [33]). The OpenRoads layer represents roads as a series of segments or 'RoadLinks', each segment starting and ending when the road changes attribution (e.g. when the road type changes definition) or at a junction [34]. There are 33,101 such segments for Birmingham. This highly granular level of detail results in many segments being short in length, with 32% of all segments being less than 50 m. As the goal was to calculate lighting information for the stretch of road on which each counter was situated, there was a danger that a particular road segment may not provide a good representation of the lighting conditions experienced by cyclists, if the segment was short. Therefore road segments were aggregated to produce longer, extended sections of road. This aggregation was done based on the 'nameTOID' variable within the OpenRoads data—those segments with the same 'nameTOID' value were combined to produce an extended section of road. This was not possible for 5,335 segments as they had no nameTOID value, and these segments were therefore ignored. The remaining 27,766 segments were combined to create 8,167

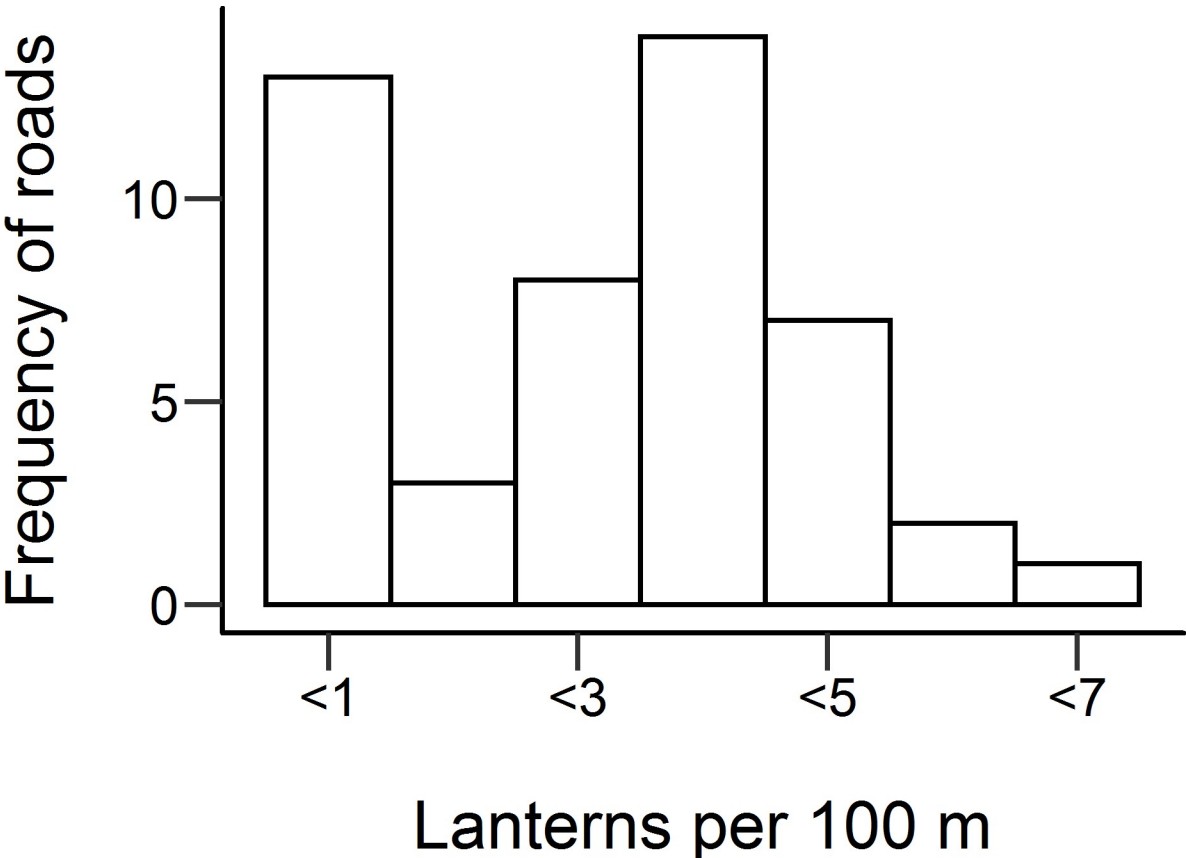

**Fig 2. Lantern density for roads in Birmingham with cycle counters located on them.** N = 48.

unique extended sections (see S4 Dataset). Just over half of the cycle counters were located on one of these sections. However, 23 counters were at locations not covered by the OpenRoads network, e.g. on cycleways or canal towpaths. A line segment was manually drawn in QGIS to represent the path that these 'off-road' counters were located on (see S5 Datset). Start and end locations were defined by the nearest junction of the path, or where it met a road within the OpenRoads network. The created segments were added to the OpenRoads layer. For simplicity, both the extended road sections created from the existing OpenRoads data and the manually-created segments at off-road locations are referred to as roads.

### Lantern density

Open source data related to public lighting in the Birmingham Local Authority district of the United Kingdom were downloaded from the Birmingham Data Factory website. The data provided spatial coordinates for lanterns within Birmingham Local Authority's inventory of public lighting, as at December 2016. This amounted to records for 94,950 lanterns (see S6 Dataset). The positions of these lanterns were plotted as an additional layer in QGIS, alongside the road network data. Lanterns were assigned to their nearest road, which could include one of the 23 manually-created segments at off-road counter locations, using the NNJoin plugin within QGIS. Lanterns that were more than 15 m away from their nearest road were judged to provide minimal impact on light levels and cycling behaviour on that road and were therefore excluded from inclusion in further analysis. This resulted in the exclusion of 12,065 lights. A

further 4,366 lights were excluded because they were assigned to road segments that had no identifying code (the 'nameTOID' variable within the OpenRoads data), and had therefore not been aggregated to form a more representative, extended stretch of road.

The density of lanterns (lanterns per 100 m) was calculated for each road with a counter located on it using Eq 2,

$$Lantern\ density_i = \frac{Lanterns_i}{Length_i/100} \tag{2}$$

Where, for the road that counter *i* is located on, Lantern density$_i$ is the number of lanterns per 100 m, Lanterns is the number of lanterns within 15 m of the road, and Length is the length of the road in metres.

The distribution of lantern densities for cycle routes is shown in Fig 2. This shows a large number of roads with lantern density less than 1 per 100 m. This includes eight roads that had no road lighting, and hence a lantern density of zero.

Lantern density is a relatively crude measure of lighting, as it does not include information about the illuminance provided by the road lighting. Using lantern density in its raw form may suggest an unrealistic scale of lighting. Cycle routes were therefore placed into three groups for further analysis, based on their lantern density: None—if the lantern density was zero; Low—if the lantern density was below the median value (2.7 lights per 100 m) for roads with lighting; and High—if the lantern density was above the median value for roads with lighting.

### Night-time aerial imagery

A series of open source raster image files providing night-time aerial photographs covering the Birmingham Local Authority area were downloaded from the UK Government's data publishing website [35]. These images were captured in March 2009 by the Environment Agency during aerial flyovers at a height of 900 m, using a colour Nikon D2X digital camera with 24 mm AF Nikkor lens and 1/100ths exposure. These 3-channel RGB images were combined to provide a complete raster layer covering the Birmingham district. This was resampled from 10 cm to 1 m pixel resolution and then saved as a single-channel greyscale image, following the procedure carried out by Hale et al [29].

Each pixel within the greyscale image had an intensity value associated with it. Previous research [29] has validated that these pixel values are highly predictive of actual illuminances measured at the location represented by the pixel ($R^2$ = 0.91). Pixel values therefore provide a reliable proxy measure of illuminance at a location. An average pixel value was calculated at each of the 48 counter locations by drawing a 15 m fixed distance buffer around the road or path segment on which the counter was situated and calculating the mean pixel value within this area.

Although the absolute value for each pixel has little practical meaning, its relative magnitude in comparison to other pixels provides an indication of relative brightness. Therefore, following the procedure used by Kuechly et al [28], a Brightness Factor (BF) was calculated for each road by dividing the mean pixel value for the road by the mean pixel value across the whole of the Birmingham district. The BF indicates the relative brightness of the road compared with the average brightness across the whole city.

## Results

All results and analysis reported here were obtained using the *R* script provided in S7 Dataset, with S2 and S3 Datasets as data inputs for this script.

## Overall effect of darkness

The yearly counts of cyclists for case and control hours by light condition are shown in
Table 3. Note that it is not appropriate to compare absolute counts across years as the number
of counters in operation in each year varied.

Odds ratios were calculated based on Eq 1 to quantify, for each year, the overall effect of
twilight and darkness on cycling rates. Counts in both control hours were summed to give a
single control count.

Odds ratios using counts in the case hour and the combined control hours are shown in Fig
3, along with their associated confidence intervals. The overall odds ratio for twilight vs day-
light when counts are summed across the whole of the 2012–2015 period is 1.12 (95% confi-
dence interval: 1.10–1.14). The overall odds ratio for darkness vs daylight is 1.32 (95%
confidence interval: 1.31–1.33). Both odds ratios are significantly greater than one (p < .001),
suggesting both darkness and twilight ambient light conditions significantly reduce the num-
ber of people cycling, compared with daylight. As the odds ratio is greater for darkness com-
pared with twilight, and the confidence intervals do not overlap, darkness is considered to
have a greater negative impact on cyclist numbers than twilight.

## Lantern density

Counter roads were categorised as having a lantern density of either None (no lighting present),
Low (below 2.7 lanterns per 100 m) or High (>2.7 lanterns per 100m). Table 4 shows median lan-
tern densities in each of these density categories, along with median BF and odds ratios for twi-
light and darkness conditions. This appears to confirm a relationship between lantern density
and relative brightness, with locations in the High density category having a larger BF than those
in the Low category, which in turn have a larger BF than those with no lighting present at all.

Odds ratios for locations with no lighting are higher than for locations that have some light-
ing present. The Tarone test of homogeneity of odds ratios [36] was used to confirm whether
differences between odds ratios for the different lantern density categories were statistically
significant. For twilight conditions, the odds ratio for locations with no lighting (OR = 1.29)
was significantly larger than locations with both low (OR = 1.11) or high (OR = 1.08) densities
of lanterns (p < .001 in both cases, Bonferroni-adjusted). The difference between odds ratios
for locations with low and high densities of lanterns was not significant however (p = 0.13).
For dark conditions, the odds ratios for locations with no lighting (2.17), low lantern density
(1.31) and high lantern density (1.20) were all significantly different from each other (p < .001
in all three cases, Bonferroni-adjusted).

## Relative brightness

Brightness Factors (BFs) were calculated for every road on which a counter was located, by
dividing the mean pixel value for the buffer area within 15 m of the road by the mean pixel

**Table 3. Overall cyclist counts by case or control hour, and light condition.**

| Light condition | Count period | 2012 | 2013 | 2014 | 2015 |
|---|---|---|---|---|---|
| Case hour in daylight | Case hour | 53027 | 97391 | 96816 | 99238 |
| | Control hours | 43307 | 78408 | 78773 | 72606 |
| Case hour in twilight | Case hour | 6778 | 11106 | 13272 | 14611 |
| | Control hours | 7076 | 8952 | 11971 | 12376 |
| Case hour in darkness | Case hour | 17277 | 32698 | 35905 | 36811 |
| | Control hours | 20204 | 35311 | 35892 | 36390 |

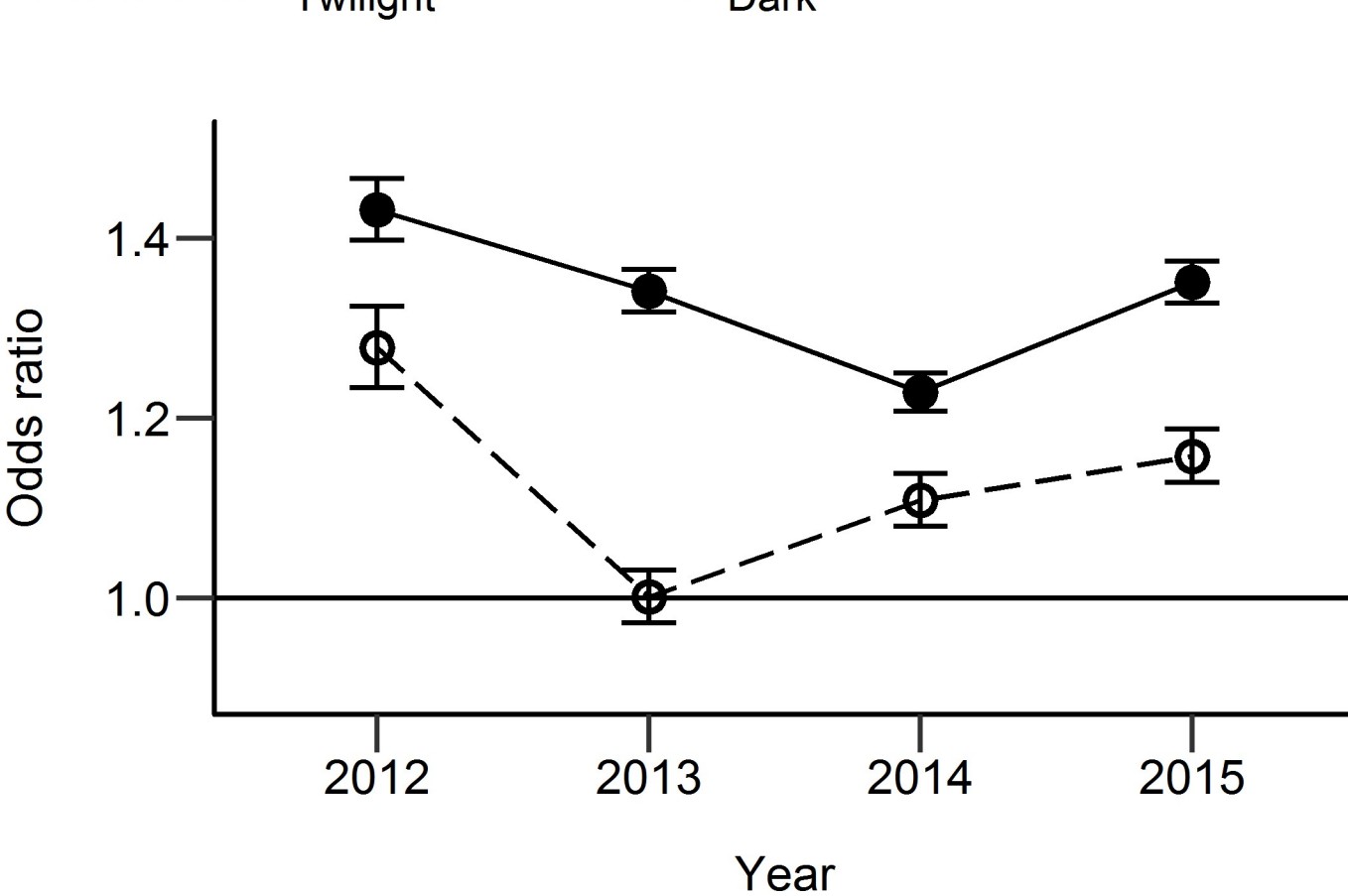

**Fig 3. Odds ratio showing effect of twilight and darkness on cyclist numbers between 2012 and 2015.** Odds ratio of 1 (indicated by horizontal bar) implies no difference in cycling rates between daylight and twilight/darkness. Odds ratio greater than 1 indicates significantly fewer cyclists after-dark compared with daylight. Error bars show 95% confidence interval.

value across the entire Birmingham district. BFs were not normally distributed, as confirmed through distribution plots and the Shapiro-Wilk test. The median BF across the 48 counter roads was 1.12 (IQR = 0.94–1.46). A Kruskal-Wallis test and post hoc Wilcoxon pairwise comparisons with Bonferroni adjustment were carried out to compare BF by the lantern density category of the road (None, Low, High). This confirmed that roads with no lighting present had significantly lower BFs (median = 0.90) than roads with low or high lantern density (medians 1.10 and 1.39, H(2) = 15.3, p < .001). There was no significant difference in BF between low and high lantern density roads (p = .31).

**Table 4. Number of counters, mean lighting statistics and mean odds ratios by lantern density category of counter roads.**

| Lantern density category | Number of counters | Median lantern density, lights per 100 m (IQR) | Median Brightness Factor (IQR) | Twilight odds ratio (95% confidence interval) | Dark odds ratio (95% confidence interval) |
|---|---|---|---|---|---|
| None | 8 | 0.00 (0.00–0.00) | 0.90 (0.90–0.94) | 1.29 (1.24–1.35) | 2.17 (2.11–2.23) |
| Low | 20 | 1.95 (0.59–2.44) | 1.10 (0.98–1.36) | 1.11 (1.08–1.13) | 1.31 (1.29–1.33) |
| High | 20 | 3.49 (3.08–4.07) | 1.39 (1.18–1.57) | 1.08 (1.06–1.10) | 1.20 (1.18–1.21) |

The twilight and dark odds ratios for each counter location were compared against the associated BF of the road at that location, to assess any relationship between the effect of ambient light conditions on cyclist counts, and the relative brightness of the road. These relationships are plotted in Figs 4 and 5. These highlight the different magnitudes of odds ratio for twilight and dark conditions. Odds ratios for both twilight and dark conditions appear to show an escarpment-plateau relationship with brightness, with odds ratios decreasing rapidly as the BF increases, but reaching an apparent asymptote as the BF increases beyond approximately 1.25. A linear regression reciprocal model, incorporating a quintic term, was used to model the non-linear relationship between odds ratio and BF. The regression lines are also plotted in Figs 4 and 5. The regression equations predicting twilight and dark odds ratios from the BF of the road are given in Eqs 3 and 4 respectively. Using these models, the BF of the road on which a counter was located was a significant predictor of the odds ratio for that counter, in both twilight and dark conditions (p < .001 in both cases).

$$OddsRatio_{Twilight} = 1.01 + 0.29\left(\frac{1}{BF^5}\right) \tag{3}$$

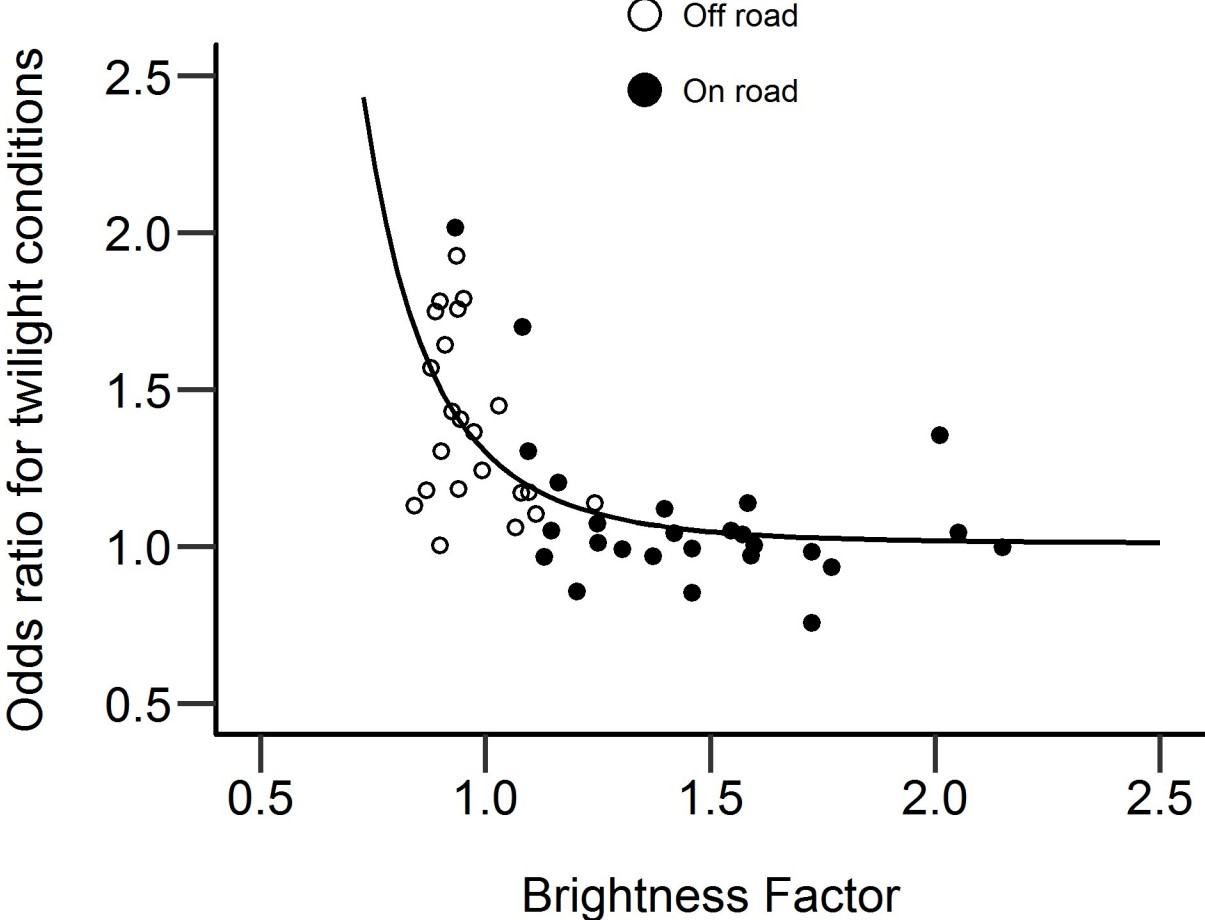

**Fig 4. Odds ratio showing effect of twilight on cyclist counts at counter locations, compared with relative brightness at the location.**
N = 48. On- and Off-road counter locations shown. Linear regression best-fit line, using a reciprocal model, is also shown.

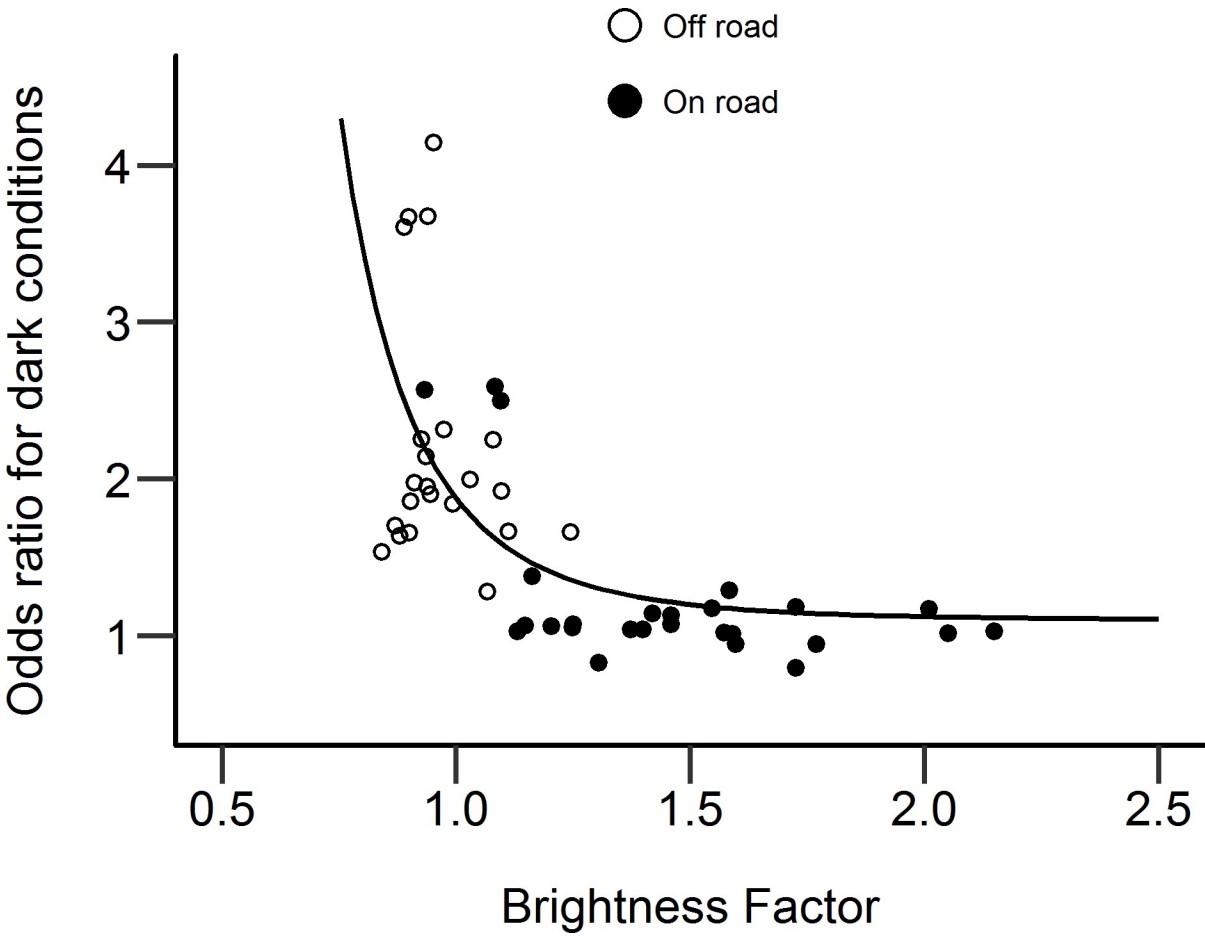

**Fig 5. Odds ratio showing effect of darkness on cyclist counts at counter locations, compared with relative brightness at the location.**
N = 48. On- and Off-road counter locations shown. Linear regression best-fit line, using a reciprocal model, is also shown.

$$OddsRatio_{Dark} = 1.10 + 0.78\left(\frac{1}{BF^5}\right) \tag{4}$$

## On-road versus off-road locations

Figs 4 and 5 suggest a link between the brightness at a location and the size of reductions in cyclists at those locations when it turns dark. These figures also show whether the counter was in an on-road or off-road location, and there is a clear distinction between counter locations in terms of the odds ratios and relative brightness at their locations. Off-road counters tend to have a low BF with a small range, but also a large range in odds ratios. On-road counters have a larger range of BF, but odds ratios are relatively low with a small range. A potential caveat to the conclusion that relative brightness is linked to the size of reductions in cyclists after-dark is the systematic variation in brightness depending on the type of location. It may be the type of location that is causing the variation in odds ratios, rather than the relative brightness.

Counters located 'off-road', e.g. on footpaths or canal towpaths, are significantly less bright (median BF = 0.94, IQR = 0.90–1.02) compared with on-road counters (median BF = 1.44, IQR = 1.21–1.60). This difference was confirmed with a Mann-Whitney U-Test (W = 23, p <

.001). Lantern density was also significantly lower at off-road locations (median = 0.44, IQR = 0.00–1.82) compared with on-road locations (median = 3.05, IQR = 2.63–3.62, confirmed with Mann-Whitney U-test, W = 58, p < .001). One possible explanation for why off-road locations might see greater reductions in cyclists when it is dark compared with on-road locations is because people may feel less safe in these locations due to fewer people being around and the locations not being observed, e.g. by passing vehicles or overlooking buildings [37].

To examine whether lighting had an effect on odds ratios independent of the location of the counter, the odds ratios of only off-road locations (N = 22) were compared by the presence or absence of lighting (note that it was not possible to perform a similar comparison for on-road locations, as all on-road locations had at least some lighting present).

The odds ratios at the off-road locations where lighting was either present or absent are shown in Fig 6. This shows the odds ratio for dark conditions at off-road locations with lighting was lower than off-road locations without lighting. This difference was confirmed as significant using the Tarone test of homogeneity of odds ratio (p < .001). The odds ratio for twilight conditions was slightly larger when lighting was absent compared with present, but this difference was marginal and was not statistically significant, according to the Tarone test (p = .078). There is evidence from these odds ratios that lighting has an effect on the reduction in cyclists

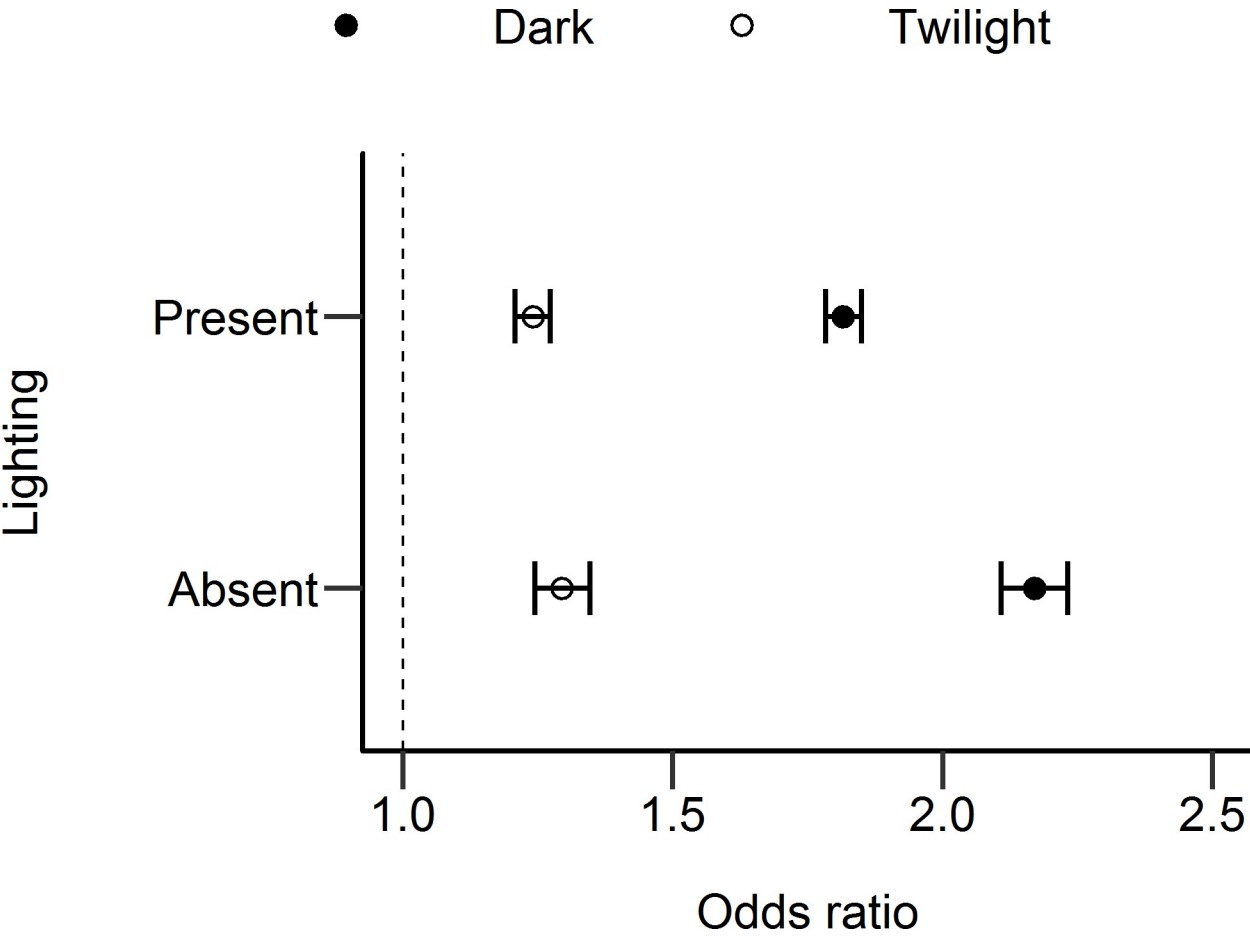

**Fig 6. Odds ratios for dark and twilight conditions, at off-road locations where road lighting was present or absent.** Error bars show 95% confidence intervals.

when it turns dark, independent of the location. If the overall effects of lantern density and relative brightness in limiting the drop in cyclist numbers after-dark were only due to the systematic variation between on-road and off-road locations, we would expect odds ratios to remain equal when the location type was held constant, regardless of whether there was lighting present or not. This was not the case however, as suggested in Fig 6.

## Discussion

This study used a case / control method to investigate the effect of darkness on cycling rates at different locations within a large urban area. We quantified this effect through the calculation of odds ratios, a measure that compares the odds an outcome will occur given exposure to some condition against the odds of the outcome without exposure to the condition [38]. The main benefit of the case / control approach used in this study is it helps control for confounding factors that may influence the outcome being investigated. In transport-related research, case / control methods and the resulting odds ratio have been used to measure effects of ambient light condition (e.g. exposure to darkness) on road traffic safety [17,18,24] and on active travel rates [14,15]. In this previous work, an odds ratio was used to quantify the effect of darkness on the variable of interest, relative to daylight, through comparison of case and control periods. The current work takes this a step further by using case and control hours to examine the link between lighting and the effect of darkness. Using the case / control design, we account for other factors that may influence cyclist numbers and therefore isolate the effect of ambient light conditions. For example, comparing cyclist numbers in daylight and darkness within a single case hour helps control for the influence of time-of-day on the number of people cycling. Comparing numbers in the case hour against numbers in the control hours helps account for seasonal factors, particularly weather conditions. The number of people cycling in the case hour when it is in daylight will undoubtedly be higher than when the case hour is in darkness, because it is in daylight during the warmer, more clement summer months and in darkness during the colder, more inclement winter months. However, an equivalent increase in numbers over the summer months will also be seen in the control hours, which are used to account for these seasonal changes in the odds ratio calculation.

A strength of the case / control research design is that it accounts for other confounding factors unrelated to light conditions that may also influence cyclist numbers. We have applied these methods in the context of cycling behaviour, but they could equally be applied in other areas where the impacts of darkness and lighting are of interest, for example road traffic safety or crime.

In previous work using odds ratios to assess effects of ambient light, daylight and darkness have effectively been treated as a binary distinction, with the light conditions either labelled as daylight or after-dark. In reality the transition between daylight and darkness, either on a specific day, or within the same hour at different times of the year, is gradual [39]. We explored the effect of this transition to darkness by calculating odds ratios for two stages of darkness: 1) Twilight, defined as the period between sunset and the end of civil twilight; and 2) Dark, defined as the period after civil twilight. Light levels during twilight are greater than darkness, and we anticipated differences in the size of odds ratios between twilight and dark conditions. This was confirmed—the overall odds ratio for twilight conditions was 1.12 (95% confidence interval: 1.10–1.14), significantly greater than 1 indicating fewer people cycled during twilight than in daylight, but also significantly less than the odds ratio for dark conditions of 1.32 (95% confidence interval: 1.31–1.33). The ordinal nature of these odds ratios (twilight odds ratio is greater than one, and dark odds ratio is greater than twilight odds ratio) supports the assumption that they reflect the impact of ambient light conditions, and not some other, unknown

variable. These odds ratios support the first two hypotheses (section 2.1), that fewer people will cycle when it is dark compared with daylight, and fewer people will cycle when it is twilight compared with daylight but the effect will not be as large as when it is fully dark. The odds ratio for darkness was not as large as has been found in previous work, with data from cyclist counters in the Arlington district of the United States producing an odds ratio of 1.67 [15], larger than that calculated from the data for Birmingham, UK as used in the current analysis. A range of factors could help explain this difference in odds ratio, such as differences in the types of counter locations, availability of other travel modes, or cycling culture. A useful area for future research would therefore be to understand the social, demographic and environmental factors that affect the decision to cycle or not when it is dark. For example, previous research has suggested gender may have an important influence on the propensity to cycle after-dark, with women more likely to be put off from cycling in darkness compared with men [19].

The odds ratio showed variation between individual years (see Fig 3). Some variation is to be expected due to random variation caused by the stochastic nature of observational data. Even if environmental and climate conditions were identical between one year and the next, we would not expect exactly the same number of people to be cycling at the same times of day and year. However, some non-random factors may also contribute to variations between years. One example relates to the operational status of individual counters. Data for a given day were only included from a counter if it was operational for both the control hours and the case hour. Despite this, being operational during parts of the year but not others could have influenced that year's overall odds ratio. For example, 2012 had the highest odds ratio for darkness of the four years. This may be due to many of the counters not being operational for all or even most of the year (see Table 2). The counters that were operational for the majority of the year tended to be in off-road locations which produce higher odds ratios (see Fig 4), resulting in a relatively high odds ratio compared with other years that had a more balanced combination of on- and off-road counters. Another possible factor contributing to variations in odds ratios between years is an interaction between climate conditions and the effect of ambient light. For example it is possible that the effect of darkness is greater when there is poor weather (colder, wetter) compared with good weather. Variations in weather conditions between the years could have led to a variable effect of darkness on cycling rates.

Although there may be random and non-random factors causing variations in the odds ratios between years, we analysed data over a four-year period and across a major city to smooth out these variations and produce a more reliable estimate of the effect of darkness and lighting on cycling rates.

We used an evening case hour to estimate the effects of darkness and twilight on cycling rates at different locations, relative to daylight. The transition over a year period between darkness, twilight, daylight, and back again, is continuous for an evening case hour. We chose not to also use a morning case hour as this transition is not continuous due to the effects of the biannual clock changes to briefly 'set back' the light condition. However, this does not rule out using a morning case hour in future similar work, and this may help to provide additional support to the findings reported here.

We calculated odds ratios to assess the effect of twilight and darkness on cyclist numbers at 48 locations across Birmingham. These were compared against two measures of lighting at each of those locations—the number of lanterns per 100 m of road / path, and the relative brightness of the road / path, assessed using night time aerial images. Odds ratios were significantly higher on roads that had no lighting, compared with roads that had some lighting present. This supports the third hypothesis, that the presence of lighting columns would reduce the negative impact of darkness and twilight on cyclist numbers. When looking at locations that

had at least some lighting present, differences between low and high lantern density locations were less obvious. The effect of twilight did not vary between these two categories of locations, although the effect of darkness was smaller for locations with high lantern density compared with low density. Odds ratios also reduced as the relative brightness of the road increased, confirming our fourth hypothesis that more brightly lit locations would show a reduced impact of darkness and twilight on cyclist numbers, compared with less brightly lit locations. However, odds ratios and relative brightness showed a nonlinear, escarpment-plateau relationship. Small initial increases in brightness led to large reductions in the odds ratio, but this effect tailed off rapidly as brightness increased further.

A possible criticism of our findings about the association between lighting and cycling rates after-dark is that better lit routes also tend to be more popular routes with higher traffic volumes due to their location, resulting in them being used by a larger number of cyclists. This assertion is not, however, supported by the evidence. Counts of cyclists in the daylight control hour (14:00–14:59) at each counter location, as an indication of the popularity of that route before lighting is factored in, are not correlated with measures of how well-lit those routes were. Total counts in the daylight control hour across the four year period (2012–2015) were not significantly correlated with lantern density ($r_s = .02$, $p = .89$) or pixel brightness ($r_s = -.19$, $p = .20$, Spearman's rank correlation used for both correlations due to data being not normally distributed). In addition, there was no correlation between the popularity of a counter location (as measured by the total count during the daylight control hour across the four year period) and the odds ratio calculated for that counter location, in both twilight ($r_s = -.11$, $p = .46$) and dark ($r_s = -.01$, $p = .97$, Spearman's rank correlation used due to data being not normally distributed). See Fig 7 for correlation plots comparing total cyclist counts during the daylight control hour against twilight and dark odds ratios for each counter location.

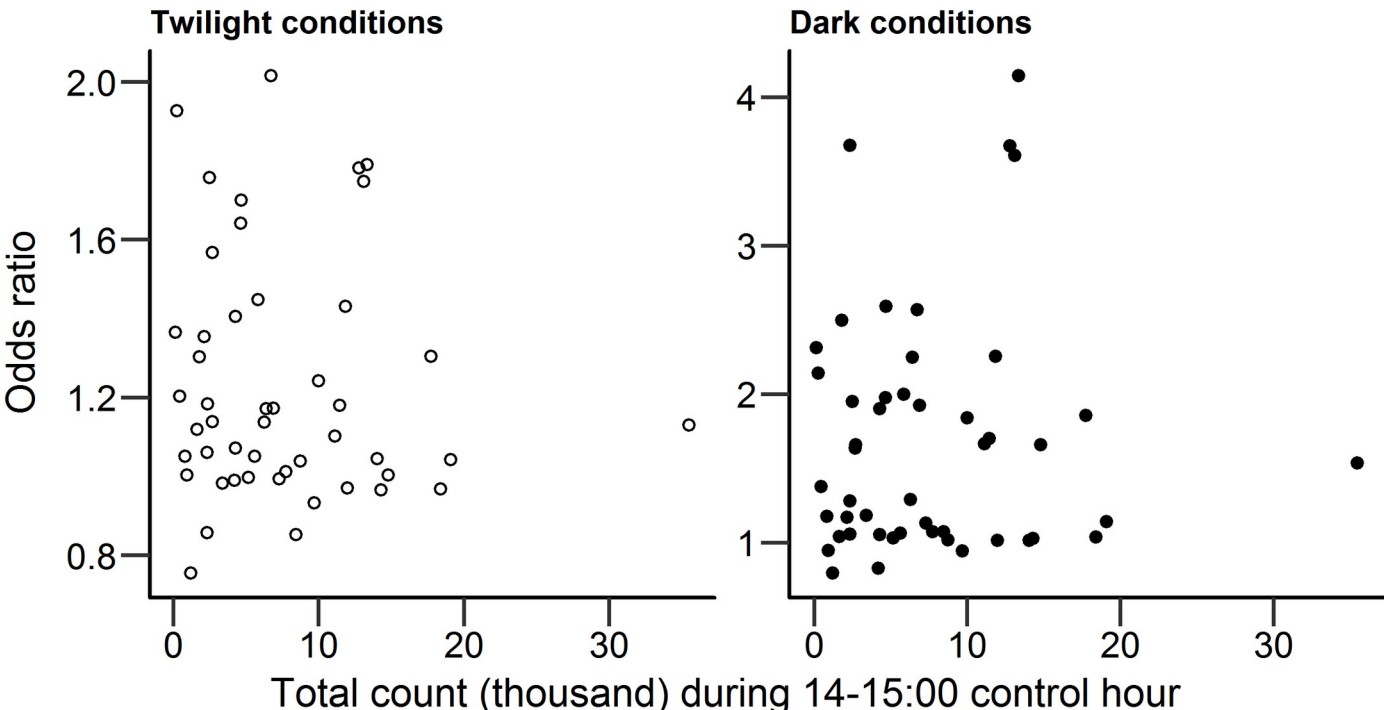

**Fig 7. Odds ratios for dark and twilight conditions compared against total counts (2012–2015) during daylight control hour (14:00–15:00), for all 48 counter locations.**

Some limitations exist with the use of the night time aerial images to indicate the brightness of the road experienced by cyclists. The images were captured in March 2009 whilst the data relating to cyclist frequencies referred to the period between 2012 and 2015. It is therefore possible that lighting conditions may have changed between the time the images were captured and the time cyclist counts were recorded, limiting the inferences that can be drawn about the relationship between light levels and cycling after-dark. It is unlikely that any changes to the lighting infrastructure will have significantly changed lighting conditions however, as roads and paths are lit according to a classification system that determines the illuminance that should be provided, based on factors such as traffic speeds, traffic volumes, presence of pedestrians and types of junctions present [26,40, 41]. These classifications are unlikely to change even if the lighting infrastructure changes, meaning any newly-installed lighting should provide similar lighting conditions to those provided previously. There is also good correspondence between values of relative brightness for a road and the density of lanterns on that road (Table 4). Lantern density was measured using data from 2016, suggesting light conditions may not have changed substantially between then and 2009, when the aerial images were captured.

A further limitation of using the night time aerial imagery to indicate brightness experienced by cyclists is it may not have captured all light that was incident on the road surface and surrounding area due to occlusion by environmental features such as overhanging trees or buildings. It is also possible the aerial images captured transient light, for example from the headlights of vehicles passing along a road that potentially exaggerated relative brightness in normal conditions. However, the association between relative brightness and the presence or absence of lighting gives some reassurance that the aerial images provided a reasonable representation of light conditions provided by fixed lighting. Previous validation work carried out by Hale et al [29], in which on-site field measurements of illuminance were compared against pixel intensities from the aerial images, also suggests the estimates of brightness from the images accurately reflected true lighting conditions and illuminance levels.

The use of lantern density as a measure of lighting conditions on a road is a relatively crude approach as it does not reflect information about the illumination provided by the lighting, such as the illuminance, its spatial distribution, and their spectra. Light spectrum may be an important factor in determining how brightly a road or urban scene is perceived, with light that has more short wavelength radiation and a higher Scotopic / Photopic luminance ratio generally being perceived as brighter in the mesopic conditions found at night [42–44]. Despite these limitations, lantern density was related to relative brightness as measured through the aerial images and potentially offers a simple yet informative basic measure of lighting on a road. Lantern density may also be linked to the uniformity of light distribution, and this is another important factor that determines the perception of an environment alongside light intensity [25].

Our approach in the analysis reported here is to examine location-based changes in cyclist numbers associated with light conditions. Whilst our case / control design helps eliminate explanations for changes in cyclist numbers that are not related to light levels, we cannot draw conclusions about possible behavioural changes that lie behind these changes in numbers. For example, we do not know whether the overall reduction in cyclist numbers associated with darkness is due to people avoiding travelling altogether, travelling at a different time of the day that is in daylight, or using a different mode of travel to cycling. Equally, we do not know whether the greater reduction in cycling rates after-dark on less well-lit routes is due to people choosing to cycle on better-lit routes instead, or deciding not to cycle at all. Further research would be useful to understand the behavioural changes and decisions made by people due to

changes in light conditions. The use of people-based (e.g. use of travel diaries) rather than location-based methods would be an appropriate approach to take [45,46].

## Conclusion

This work confirms that the number of people cycling when dark is significantly lower than in daylight, after time-of-day and seasonal factors are controlled for. The research presented in this paper supports the view that road lighting reduces this negative effect of darkness on cycling rates and encourages people to continue cycling after dark, with associated benefits for health and the environment. These findings suggest that use of lighting on cycle routes and the light conditions provided should be an important consideration for local transport planners as they seek to promote cycling activity in line with strategic policy, such as the Cycling and Walking Investment Strategy in the United Kingdom [7].

Odds ratios using 'case' and 'control' hours are a useful method for examining the effect of ambient light conditions on travel behaviour. We have used them to extend previous work by demonstrating the potential benefits of lighting in negating the effects of darkness on cycling and encouraging more people to cycle. The presence of lighting on a road can have a positive effect on cyclist numbers after-dark. However, a high density of lights may not be required to obtain the beneficial effects of lighting. Similarly, only a small increase in brightness can have a big impact on cycling rates after-dark, and further increases in brightness may not provide major benefit in terms of encouraging more cycling. These findings suggest only a minimal level of lighting is required to achieve beneficial effects for cycling rates after-dark. One limitation of the current work is that the lighting measures used, lantern density and relative brightness, do not provide information about actual illuminance levels. Our findings can be further validated and extended by assessing the relationship between cycling after-dark and actual illuminance values on roads, as measured using standard lighting practice methods [41]. This can support street lighting practitioners to establish light levels that encourage active travel. Many Local Authorities in the UK and elsewhere are currently making changes to their lighting portfolio, for example by installing new LED light sources. The methods used in the current analysis would lend themselves to assessing the impacts of such changes to lighting, in a before-after evaluation.

It is important to use lighting efficiently and not waste energy through unnecessary lighting. The carbon trade-off between more lighting and increased cycling may lie more in favour of increased lighting though. A 50 W LED street light will use 0.55 kWh per day (assuming an average 11 hour burn time each day). This equates to 0.14 kgCO$^2$e per day, per light, based on UK greenhouse gas conversion factors for producing electricity [47]. Cars are estimated to produce 0.27 kgCO$^2$e per passenger-kilometre, for trips equivalent to those taken by cyclists [48]. Adding 10 new street lights to a cycle path would therefore only need 6 passenger-kilometres to be shifted from driving to cycling in order to offset the additional carbon emissions they produce. This is equivalent to just one additional person commuting by bike rather than car, in order offset the carbon emissions from the additional lighting. Carbon trade-offs are not the only consideration though, as the health benefits of more cycling and the potential ecological impacts of more lighting also need considering.

Whilst our results suggest only a minimal amount of lighting may be required to encourage more cycling after-dark, it is also worth noting there may be other requirements for lighting in order to keep cyclists safe. Lighting can help cyclists see and avoid potential hazards such as pot holes [23], and can make cyclists more visible to drivers thereby reducing road traffic collisions involving cyclists [49]. Encouraging more cycling is not the only consideration for lighting therefore, and lighting should be considered alongside other interventions, such as high

quality cycleways and 'liveable streets' [50,51], to make cycling a safe and enjoyable activity for all. This may require varying degrees of lighting, depending on its purpose.

## Supporting information

**S1 Dataset. Shapefile showing locations of the 48 cycle counters in Birmingham, UK.**
(ZIP)

**S2 Dataset. CSV file with yearly count data per counter, by condition (case or control hour) and light condition during the case hour (day or dark).**
(CSV)

**S3 Dataset. CSV file with aggregated count data per counter, calculated lantern density and brightness factor at each location.**
(CSV)

**S4 Dataset. CSV file with details of combined road segments.**
(CSV)

**S5 Dataset. Shapefile showing vectors created for non-road segments at off-road counter locations.**
(ZIP)

**S6 Dataset. CSV file with data about road lighting locations in Birmingham, UK.**
(CSV)

**S7 Dataset. R script for all results and analysis reported in this paper.**
(R)

## Author Contributions

**Conceptualization:** Jim Uttley, Steve Fotios, Robin Lovelace.

**Data curation:** Jim Uttley.

**Formal analysis:** Jim Uttley.

**Funding acquisition:** Jim Uttley, Steve Fotios.

**Methodology:** Jim Uttley, Robin Lovelace.

**Supervision:** Steve Fotios.

**Visualization:** Jim Uttley.

**Writing – original draft:** Jim Uttley.

**Writing – review & editing:** Steve Fotios, Robin Lovelace.

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
