## [Decision Letter · Decision Letter 0]

5 Mar 2020

PONE-D-20-02887

Road lighting density and brightness linked with increased cycling rates after-dark

PLOS ONE

Dear Dr. Uttley,

Thank you for submitting your manuscript to PLOS ONE. After careful consideration, we feel that it has merit but does not fully meet PLOS ONE’s publication criteria as it currently stands. Therefore, we invite you to submit a revised version of the manuscript that addresses the points raised during the review process.

Please systematically address each reviewer's comments.

We would appreciate receiving your revised manuscript by Apr 19 2020 11:59PM. To enhance the reproducibility of your results, we recommend that if applicable you deposit your laboratory protocols in protocols.io, where a protocol can be assigned its own identifier (DOI) such that it can be cited independently in the future. For instructions see: http://journals.plos.org/plosone/s/submission-guidelines#loc-laboratory-protocols

We look forward to receiving your revised manuscript.

Kind regards,

Jake Olivier, Ph.D.

Academic Editor

PLOS ONE

Journal Requirements:

2. We note that [Figures 1 and 3] in your submission contain [map/satellite] images which may be copyrighted. All PLOS content is published under the Creative Commons Attribution License (CC BY 4.0), which means that the manuscript, images, and Supporting Information files will be freely available online, and any third party is permitted to access, download, copy, distribute, and use these materials in any way, even commercially, with proper attribution. For these reasons, we cannot publish previously copyrighted maps or satellite images created using proprietary data, such as Google software (Google Maps, Street View, and Earth). For more information, see our copyright guidelines: http://journals.plos.org/plosone/s/licenses-and-copyright.

1.     You may seek permission from the original copyright holder of Figures [1 and 3] to publish the content specifically under the CC BY 4.0 license.  

Reviewers' comments:

Reviewer's Responses to Questions

**Comments to the Author**

1. Is the manuscript technically sound, and do the data support the conclusions?

Reviewer #1: Yes

Reviewer #2: Yes

2. Has the statistical analysis been performed appropriately and rigorously? 

Reviewer #1: Yes

Reviewer #2: Yes

3. Have the authors made all data underlying the findings in their manuscript fully available?

Reviewer #1: Yes

Reviewer #2: Yes

4. Is the manuscript presented in an intelligible fashion and written in standard English?

Reviewer #1: Yes

Reviewer #2: Yes

5. Review Comments to the Author

Reviewer #1: The paper explores whether road lighting can reduce the negative impact of

darkness on cycling rates. The issue studied is relevant, the methods are generally adequate and the findings have the potential to inform policy in this area. However, there are a number of issues that need to be addressed to strengthen the manuscript.

The aim/objectives need to be stated clearly at the end of the introduction. Hypotheses ought to be listed at the end of introduction and not in the methods.

Similarly, the statements about the use of more precise quantitative definitions

of lighting (lantern density and relative brightness) at the end of the introduction need to be moved to the methods.

In the methods, justification for choosing (22:00-22:59) as opposed for example to (21:00-21:59), which is also I assume dark throughout the year, is needed.

A segmentation of the analysis by areas/roads that are popular with cyclists vs those that are less popular would be useful to partly address at least one of the limitations discussed by the authors.

Reviewer #2: This is a clear and well-written paper. My initial questions about potential other factors which could influence cycling at night and be correlated with lighting levels were satisfactorily addressed in the paper. I assume that the effects of daylight saving (and any potential changes across years in start/end dates) were dealt with.

I wasn't sure why only one case hour was chosen. Why wasn't a morning case hour used as well? If this was done, and similar results were found for the two case hours, that would be supportive evidence.

It would be interesting for the authors to discuss (or note) the issue of whether the low level of lighting needed to encourage riding in the dark would be sufficient to allow safe cycling. It might not be a good idea to encourage riding that was objectively high risk.

A second minor issue is that the authors make the point that cycling is good for climate because it saves energy. Is there a trade-off, then, if more energy needs to be used for lighting in order to increase cycling?

6. PLOS authors have the option to publish the peer review history of their article (what does this mean?). If published, this will include your full peer review and any attached files.

Reviewer #1: No

Reviewer #2: Yes: Narelle Haworth

---

## [Author Response · Author response to Decision Letter 0]

25 Mar 2020

Please see 'Response to Reviewers' document

---

## [Decision Letter · Decision Letter 1]

29 Apr 2020

Road lighting density and brightness linked with increased cycling rates after-dark

PONE-D-20-02887R1

Dear Dr. Uttley,

We are pleased to inform you that your manuscript has been judged scientifically suitable for publication and will be formally accepted for publication once it complies with all outstanding technical requirements.

With kind regards,

Quan Yuan, Ph.D.

Academic Editor

PLOS ONE

Additional Editor Comments (optional):

Reviewers' comments:

Reviewer's Responses to Questions

**Comments to the Author**

1. If the authors have adequately addressed your comments raised in a previous round of review and you feel that this manuscript is now acceptable for publication, you may indicate that here to bypass the “Comments to the Author” section, enter your conflict of interest statement in the “Confidential to Editor” section, and submit your "Accept" recommendation.

Reviewer #2: All comments have been addressed

2. Is the manuscript technically sound, and do the data support the conclusions?

Reviewer #2: (No Response)

3. Has the statistical analysis been performed appropriately and rigorously? 

Reviewer #2: (No Response)

4. Have the authors made all data underlying the findings in their manuscript fully available?

Reviewer #2: (No Response)

5. Is the manuscript presented in an intelligible fashion and written in standard English?

Reviewer #2: (No Response)

6. Review Comments to the Author

Reviewer #2: (No Response)

7. PLOS authors have the option to publish the peer review history of their article (what does this mean?). If published, this will include your full peer review and any attached files.

Reviewer #2: Yes: Narelle Haworth